# Inductive Biases for Object-Centric Representations of Complex Textures

## Abstract

Understanding which inductive biases could be useful for unsupervised models to learn object-centric representations of natural scenes is challenging. Here, we use neural style transfer to generate datasets where objects have complex textures. Our main finding is that a model that reconstructs both the shape and visual appearance of each object from its representation achieves correct separation of the objects and learns useful object representations more reliably.

## 1 Introduction

A core motivation for object-centric learning is that humans interpret complex environments such as natural scenes as the composition of distinct interacting objects. Evidence for this claim can be found in cognitive psychology and neuroscience (Spelke, 1990; Téglás et al., 2011; Wagemans, 2015), particularly in infants (Dehaene, 2020, ch. 3). Additionally, these concepts have already been successfully applied to various fields, from reinforcement learning (Vinyals et al., 2019; OpenAI et al., 2019) to physical modelling (Battaglia et al., 2016; Sanchez-Gonzalez et al., 2020). Current object-centric learning approaches try to merge the positives of connectionist and symbolic methods by representing each object with a distinct vector (Greff et al., 2020). The problem of separation of the objects becomes central for unsupervised methods because, with no additional information other than the data itself, learning to isolate objects in the input can arguably be a challenging task.

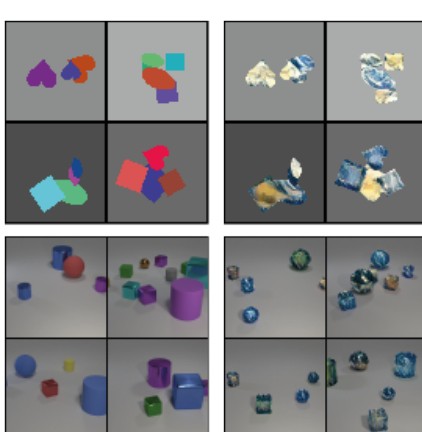

Figure 1: Left: samples from the original datasets. Right: samples from the same datasets with neural style transfer.

Several methods have been proposed to provide inductive biases to achieve this objective (the most relevant for this work are Burgess et al. (2019); Engelcke et al. (2020b); Locatello et al. (2020); Engelcke et al. (2020a); Greff et al. (2019); Kipf et al. (2021b); Engelcke et al. (2021), but see Appendix for additional references). However, they are typically tested on relatively simple datasets where the objects show little variability in their texture, often being monochromatic. This characteristic may allow object-centric models to successfully separate objects by relying solely on low-level characteristics, such as color (Greff et al., 2019), over more desirable high-level ones, such as shape or even behavior in videos (Kipf et al., 2021a).

Little research has been done in the direction of natural objects (Engelcke et al., 2021; Kipf et al., 2021a), as datasets with such characteristics often do not feature samples that provide exhaustive knowledge of the underlying factors of variation, which are very rich in natural scenes. For this reason, unsupervised methods struggle to learn object-centric representations and little can be understood regarding the reasons why they are failing (Greff et al., 2019, Sec. 5).

*In this paper, we conduct a systematic experimental study on the inductive biases necessary to learn object-centric representations when objects have complex textures.* We make practical choices to obtain significant and interpretable results. We focus on static images and use neural style transfer (Gatys et al., 2016) to apply complex textures to the objects of the Multi-dSprites (Kabra et al., 2019) and CLEVR (Johnson et al., 2017) datasets. The increase in complexity is, therefore, controlled: On

the one hand, we still have all of the advantages of a procedurally generated dataset, with knowledge over the characteristics of each object, thus avoiding the above-mentioned pitfalls of natural datasets. On the other hand, we present a much more challenging task for the models than the type of data commonly used in unsupervised object-centric learning research. We investigate MONet (Burgess et al., 2019) and Slot Attention (Locatello et al., 2020), two popular and successful approaches for unsupervised object-centric learning. Both are slot-based autoencoder models that learn to represent objects separately and in a common format. MONet has *two* separate components: a recurrent attention network that segments the input into objects, and a variational autoencoder (VAE) (Kingma & Welling, 2014; Rezende et al., 2014) that separately learns a representation for each object. The Slot Attention autoencoder obtains object representations by applying Slot Attention to a convolutional embedding of the input, and then decodes each object representation into RGB color channels and an alpha mask, thus performing separation and reconstruction with a *single* component.

For this study, we posit two desiderata for object-centric models, adapted from Dittadi et al. (2021):

**Desideratum 1.** *Object separation and reconstruction.* The models should have the ability to accurately separate and reconstruct the objects that are present in the input, even those with complex textures. For the models considered here, this means that they should be able to correctly segment the objects and reproduce their properties in the reconstruction, including their texture.

**Desideratum 2.** *Object representation.* The models should capture and represent the fundamental properties of each object present in the input. When ground-truth properties are available for the objects, this can be evaluated via a downstream prediction task.

We summarize our **main findings** as follows:

1. Models that use a single mechanism to reconstruct both the shape and the visual appearance of the objects in the input appear to be less prone to what we call *hyper-segmentation* (see Section 3.1). This suggests that, moving forward, such architectures are to be preferred.

2. Hyper-segmentation of the input leads to the inability of the model to successfully encode the characteristics of the elements present in the input. Separation is a reliable indicator of the quality of the representations.

3. The *representation bottleneck* is not sufficient to regulate a model's ability to segment the input. Tuning of hyperparameters such as encoder and decoder capacity is necessary.

In the following, we will discuss our findings and lay out practical suggestions for researchers in the object-centric learning field to push towards making models work on natural images.

## 2 METHOD

In this section, we outline the elements of our study, highlighting the reasons behind our choices.

**Datasets.** Similarly to Dittadi et al. (2021), we use neural style transfer (Gatys et al., 2016) to increase the complexity of the texture of the objects in the Multi-dSprites and CLEVR datasets (see Appendix for details). This allows for textures that have high variability but are still correlated with the shape of the object, as opposed to preset patterns as done in Greff et al. (2019) or completely random ones. We apply neural style transfer to the entire image and then select the objects using the ground-truth segmentation masks (see Fig. 1). Keeping the background simple allows for easier interpretation of the models' performance.

**Models.** The models we study are MONet (Burgess et al., 2019) and Slot Attention (Locatello et al., 2020), that approach the problem of separation in two distinct ways, as mentioned in Section 1. MONet uses a recurrent attention module to compute the shape of the objects, and only later is this combined with the visual appearance computed by the VAE from the object representations. Instead, Slot Attention incorporates everything into a single component, with the shape and visual appearance of each object reconstructed from the respective object representation.

**Evaluation.** Following the two desiderata in Section 1, we separately focus on the *separation*, *reconstruction*, and *representation* performance of the models. *Separation* is measured by the Adjusted Rand Index (ARI) (Hubert & Arabi, 1985), which quantifies the similarity between two partitions of a set. The ARI is 0 when the two partitions are random and 1 when they are identical up to

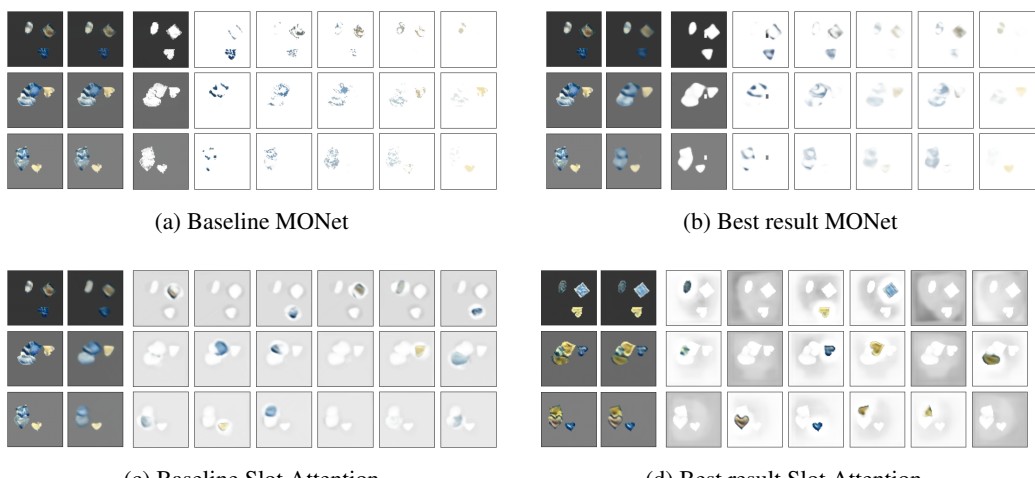

(a) Baseline MONet

(b) Best result MONet

(c) Baseline Slot Attention

(d) Best result Slot Attention

Figure 2: Qualitative results for the reconstruction and separation performance of the models in the comparative study (from validation set). From left to right in each subfigure: original input, final reconstruction, and product of the reconstruction and mask for each of the six slots. The improved architecture for Slot Attention splits fewer objects. MONet still fails to separate objects correctly but segments them over a smaller number of slots.

a permutation of the labels. *Reconstruction* is measured using the Mean Squared Error (MSE) between input and reconstructed images. *Representation* is measured by the performance of a simple downstream model trained to predict the properties of each object using only the object representations as inputs. Following previous literature (Locatello et al., 2020; Dittadi et al., 2021), we match the pairs (*object representation*, *object*) such that the overall loss is minimized.

**Performance studies.** We first establish the performance of the models on the style transfer datasets when using the parameters specified in the original papers (for MONet we use those from Greff et al. (2019)). We then vary parameters and architectures in an attempt to improve performance. In MONet, we modify the attention module, the latent space size, the number of channels in the encoder and decoder of the VAE, and some parameters in the training objective (details in Appendix). In Slot Attention, we increase the number of layers and channels in both encoder and decoder and increase the size of the latent space. For both models, we investigate how the latent space size alone affects performance. We use multiple random seeds to account for variability in performance.

## 3 EXPERIMENTS

In this section, we present and discuss the experimental results of our study. We first look at how different architectural biases affect object separation. Then, we investigate representation performance with a downstream property prediction task. Finally, we study how the latent space size alone affects object separation and reconstruction quality.

### 3.1 ARCHITECTURAL BIASES

Fig. 2 shows the qualitative performance of a selection of models. The MONet baseline (Fig. 2a) shows a rather peculiar behavior: the scene is segmented primarily according to color so that each slot represents fragments of multiple objects that share the same color. We call this behavior *hypersegmentation*. On the other hand, the Slot Attention baseline (Fig. 2c) produces blurred reconstructions but avoids hyper-segmentation. Here, some objects are still split across more than one slot but, unlike in MONet, we do not observe multiple objects that are far apart in the scene being (partially) modeled by the same slot. We observe this quantitatively in Fig. 3: compared to the Slot Attention baseline, the MONet baseline has a significantly worse ARI score but a considerably better MSE.

These observations can guide our search for better model parameters. Slot Attention is blurring away the small details of the texture and instead focuses on the shape to separate them. MONet, instead, achieves good reconstructions but does so by using the attention module to select pixels that

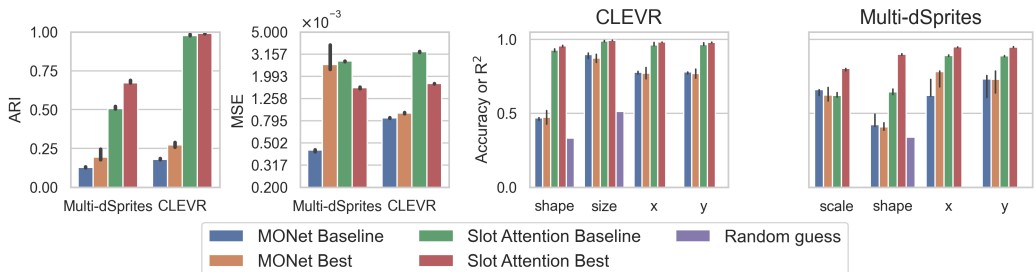

Figure 3: Bar plots of the median performance of the different seeds trained for each of the indicated models (error bars denote 95% confidence intervals). Left: ARI (↑) and MSE (↓) for each dataset and model. Right: performance of the downstream model on each feature of the objects. Accuracy is used for categorical features and $R^2$ for numerical features. A random guess baseline is shown in purple.

share the same color, as opposed to entire objects. Therefore, for MONet we attempt to sacrifice some reconstruction quality in exchange for better object separation. For Slot Attention, we investigate whether improving the reconstructions negatively affects object separation. We refer to the Appendix for further details on the hyperparameter search for MONet and Slot Attention.

The best results obtained by MONet (Fig. 2b) show persisting hyper-segmentation even though the reconstructions are now blurred. For the best Slot Attention model (Fig. 2d), we observe that the quality of reconstructions has improved, and that the model more often represents an entire object in a single slot. Although the ARI for MONet was improved, we could not solve the separation problem, while Slot Attention shows a significant improvement both in terms of ARI and MSE (see Fig. 3). Note that, although the ARI has improved from the baseline, the best Slot Attention model cannot achieve a very high ARI score in Multi-dSprites, unlike in CLEVR. The likely reason is that, when a significant portion of an object is occluded by another, the shape is being altered significantly and the edges of objects are not clear. Therefore, two explanations of the same scene can still be reasonable while not corresponding to the ground truth. This extreme overlap never happens in CLEVR because of how the dataset was generated, which explains the difference in ARI score.

Overall, even when MONet partially sacrifices reconstruction quality and begins to blur away the details, hyper-segmentation is still present, as evidenced by our qualitative and quantitative analyses. This suggests that the separation problem in MONet is not directly caused by the training objective, but rather by its architectural biases. Indeed, increasing the reconstruction performance of Slot Attention has, instead, yielded both better separation and more detailed reconstructions, showing how generating shape and appearance using a single module is a more favorable inductive bias to learn representations of objects with complex textures.

## 3.2 REPRESENTATION PERFORMANCE

To understand the role of separation when learning object-centric representations, we explore the performance of a downstream property prediction task. A model predicts the feature of each object starting from its representation (see Appendix for details). From Fig. 3), we observe how MONet fails to capture some of the properties in the representations and consistently performs worse than Slot Attention, for both the baseline and the improved versions. This suggests that, as highlighted in Dittadi et al. (2021), a model that is not capable of correctly separating objects will also fail to accurately represent them. The trend is also clear from Fig. 4, which shows that a higher ARI score strongly correlates with an increased performance of the downstream model on all object properties. The correlation with MSE is much weaker, which highlights how *strong visual reconstruction performance is not the ultimate indicator for good object representations*. This result does not contradict previous findings (Dittadi et al., 2021) as here the properties we expect the downstream model to predict have little to do with the texture of the object and, therefore, the model can have poorer reconstructions while still obtaining useful representations.

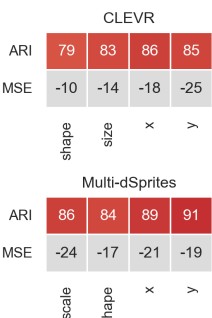

Figure 4: Correlation between downstream performance and the ARI and MSE over all models.

Figure 5: ARI (↑) and MSE (↓) performance for different latent space sizes. For each size two seeds are used.

### 3.3 REPRESENTATION BOTTLENECK

Object representations are typically obtained by using a low-dimensional latent space for each object. This presents itself as a bottleneck in the model, which we term *representation bottleneck* (see Appendix). Here, we investigate how the size of this bottleneck affects performance. We observe from Fig. 5 that the MSE improves for both MONet and Slot Attention when the latent space size increases. However, for MONet this comes with a decrease in separation performance. For Slot Attention, when the latent space reaches a critical size (256 in CLEVR and 512 in Multi-dSprites), the performance degrades, and the variability across seeds increases drastically, showing how this is likely caused by problems during the training procedure. The increase in latent space size arguably increases the capacity of the model, but it does not prove to be enough to significantly improve the separation and reconstruction performance. We could obtain considerable improvements only by changing the architectures.

## 4 RELATED WORK

Greff et al. (2019) train IODINE on Textured MNIST and ImageNet, and observe that the model separates the image primarily according to color when the input is complex, which is in agreement with our findings. GENESIS-V2 (Engelcke et al., 2021) was trained on the real-world robot manipulation datasets Sketchy and APC. However, the authors do not explore the mechanism behind the performance of the models they tested, and do not attempt to optimize the architectures. Engelcke et al. (2020a) study the inductive biases in object-centric learning for real-world images and investigate the reasons why the models are unable to separate objects in different slots. However, the focus is only on one specific model and on traditional synthetic datasets, while in our work we study hyper-segmentation on datasets where objects have complex textures. In the video domain, Kipf et al. (2021b) include evaluations on a dataset with complex textures, but train their model to predict optical flow rather than reconstruct the input.

## 5 CONCLUSIONS

We have tackled the problem of understanding which inductive biases may be most useful for slot-based unsupervised models to obtain good object-centric representations of objects with complex textures. We found that having a single module that reconstructs both shape and visual appearance is a suitable approach for more consistent separation by avoiding what we call hyper-segmentation. Therefore, our recommendation is that models should have separation as integral part of the representation process. Additionally, we showed that separation strongly correlates with the quality of the representations. Finally, we observed that the representation bottleneck is not a sufficient inductive bias, as increasing the latent space size can be counterproductive unless the model is already separating the input correctly.

We limited our study to two models based on instance slots and sequential slots. Although they have been shown in the literature to be the ones that are most successful (Dittadi et al., 2021), it would be interesting and natural to extend our study and explore if the same holds for other models that approach the problem in a similar way, such as GENESIS, IODINE, and GENESIS-V2, as well as methods based on spatial slots, such as SPACE. Another interesting avenue for future work is to use different datasets and to have objects with mixed texture complexity, as this could require different model capacities to achieve separation (Engelcke et al., 2020a).

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

# A NOTES ON TERMINOLOGY, DEFINITIONS, AND REFERENCES

## A.1 SOME OBSERVATIONS REGARDING THE NOTATION USED

The term *representation bottleneck* should not be confused with the *reconstruction bottleneck* introduced by Engelcke et al. (2020a). The representation bottleneck refers to the small size of the latent space, instead, the reconstruction bottleneck refers to how easy it is for the model to reconstruct something. In Engelcke et al. (2020a), the reconstruction bottleneck is posited to be the reason behind the models not being able to separate distinct objects in different scenes, and instead reconstructing the entire image with just one single object representation, rendering the model useless.

Often, in the paper, we refer to *object-centric representations* and *slots* as synonyms, although this is only true for slot-based models.

The term *hyper-segmentation* refers to when a model splits the input into different slots with little to no regard to high-level characteristics of the input, such as shape, and instead just uses low-level characteristics, primarily color. This results in small clusters of pixels, all with similar color, in a slot, which means that several objects can share the same object representation and at the same time be represented in multiple object representations. This phenomenon is distinct from *over-segmentation*, where multiple parts of different objects will not appear in a single slot. Examples are seen in the main text of the paper, where MONet is hyper-segmenting the input, while Slot Attention is over-segmenting it at times.

## A.2 ADDITIONAL REFERENCES FOR OBJECT-CENTRIC MODELS

Other relevant work in the object-centric literature is the following: Deng et al. (2021); Mnih et al. (2014); Nash et al. (2017); Gregor et al. (2015); Eslami et al. (2016); Kosiorek et al. (2018); Stelzner et al. (2019); Crawford & Pineau (2019); Lin et al. (2020); Yuan et al. (2019); Dittadi & Winther (2019); Weis et al. (2020); von Kügelgen et al. (2020); Greff et al. (2017); Jiang* et al. (2020); Chen et al. (2020); Crawford & Pineau (2020). These papers primarily present new models or approaches for object-centric learning.

The paper Greff et al. (2020) provides an overall summary and categorization of most of the recent models and overall approaches to the problem of object-centric representations.

# B DATASETS

The original versions of both datasets are taken from Kabra et al. (2019).

**CLEVR** The CLEVR dataset consists of 3D objects placed on a gray background at different distances from the camera. Overlap between objects is kept to a minimum. There are spheres, cylinders, and cubes of eight different colors. The objects can be metallic of opaque in the material. There is a big and small variant of each object and they can be placed in several different orientations.

We use the variant of the dataset that has no more than 6 objects in it, as was done in previous object-centric research. The total number of samples in the training dataset is 49483, and we leave 2000 samples for validation and testing.

**Multi-dSprites** The Multi-dSprites dataset places several 2D objects on a grayscale background. The objects can be a square, an ellipse, or a heart. They can have any RGB color, orientation, and different levels of overlap.

Here, we use 90000 samples for the training, and 5000 for validation and testing.

**Neural Style Transfer** Neural Style Transfer was applied in its most basic form, which can be found in a Pytorch tutorial (Jacq, 2021) minus a few additions to make running it on several datasets easier. We opted to use *The Starry Night* by Dutch painter *Vincent Van Gogh* as a reference style image (we used the photo from Wikimedia Commons, which is in the public domain). We experimented with several parameters, and we noticed a lot of variability between runs and a more pleasant result from the most basic implementation of the algorithm. After we obtained the neural

style transfer version of each sample in the dataset we applied the original segmentation masks to obtain a version of the dataset where only the foreground objects have a complex texture.

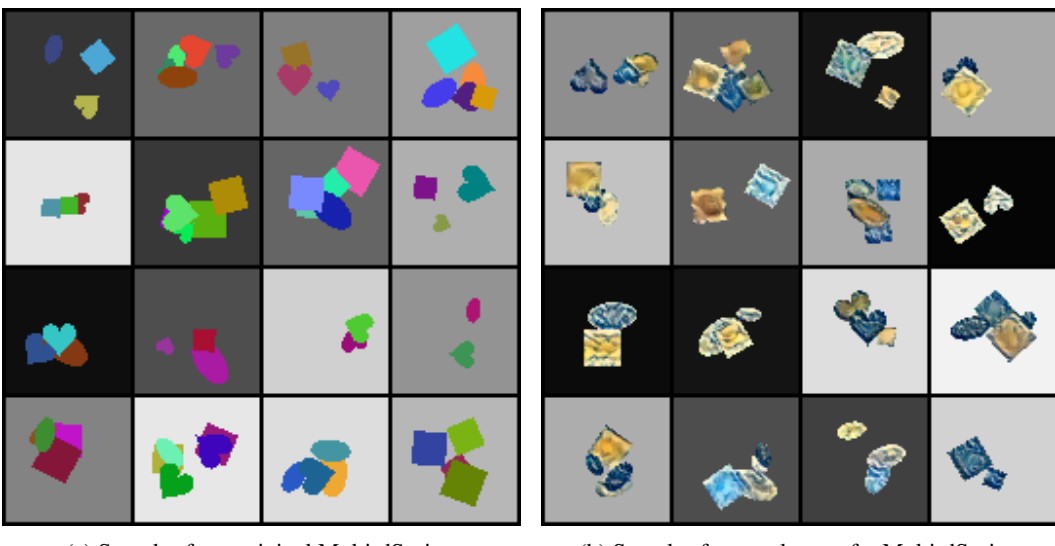

(a) Samples from original Multi-dSprites          (b) Samples from style transfer Multi-dSprites

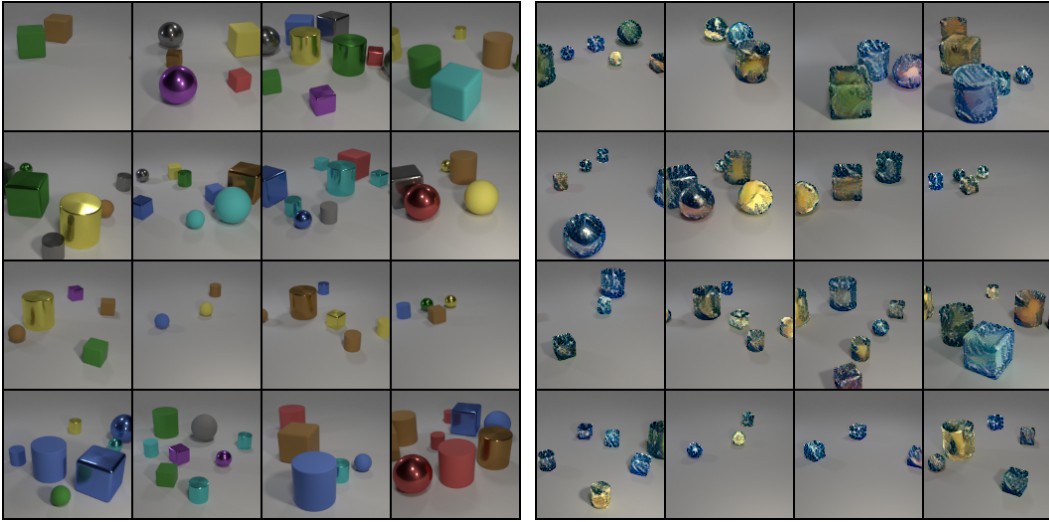

(c) Samples from original CLEVR          (d) Samples from style transfer CLEVR

Figure 6: Samples from the original and neural style transfer datasets.

## C  EVALUATION

### C.1  DOWNSTREAM FEATURE PREDICTION TASK

The feature prediction task is the same as the one used in Dittadi et al. (2021). It uses a simple MLP with one hidden layer having 256 neurons and enough outputs to predict all of the features of an object. The input to the model is the object representation of a single object and the output will be the predicted features of that object. For those outputs that are numerical, an MSE loss is used, while for the categorical ones, a cross-entropy loss is used. The loss is calculated for all pairs of objects and object representations. This creates a loss matrix to which the Hungarian matching algorithm is applied to find the pairs that minimize the sum of the loss. The model is then trained by minimizing the sum of the loss given by the selected pairs.

### C.2  ARI AND MSE

We use the traditional definition of Adjusted Rand Index and Mean Square Error.

The ARI is computed on the foreground objects and is meant to measure the similarity between two partitions of the same set. The *adjusted* part of the name comes from the fact that the Rand Index has been normalized according to a null hypothesis to give 0 when the partitions are random and 1 when they coincide.

The MSE is computed between each channel in each pixel of the image.

## D  IMPLEMENTATION OF THE MODELS

The models were re-implemented in PyTorch (Paszke et al., 2019) and run on both NVIDIA A100 and NVIDIA TitanRTX GPUs. The total approximate training time to reproduce this study is 180 GPU days.

## E  HYPERPARAMETER SEARCHES

### E.1  BASELINES

The baselines were obtained by training the models on the two datasets with 3 different seeds. The parameters are taken from the original papers, but for MONet we use different values for the foreground and background sigma, as suggested in Greff et al. (2019). We stopped the training for all runs in our study, even the ones described later, to 300k steps.

### E.2  IMPROVING MONET

Starting from the baseline results, we first tried some parameters manually to try and get an idea regarding what was the effect of each parameter on the performance.

We then performed a hyper-parameter search to find the best parameters for MONet. We did a full search, resulting in 36 runs. Because of the high number of runs. The parameters are listed in Table 1. Those that are not listed were kept the same. The parameters foreground sigma and background sigma are changed in pairs (so when making a run, the first value for each is used, resulting in a factor of 2 more runs and not 4 times more runs to test these parameters).

Some analysis on the results of these models can be seen in Fig. 7, where we can see how the parameters have little to no impact on the overall performance of the model. What proved to be most effective was reducing the number of skip connections in the U-Net and using a small sigma for the loss function. However, these results are not very conclusive, as a small number of skip connections is actually just increasing the ARI slightly by reconstructing bigger patches of objects in the slots and not actually separating them correctly.

| Parameter | Value(s) |
|---|---|
| foreground sigma | $0.05, 0.5$ |
| background sigma | $0.03, 0.3$ |
| gamma | $1, 5, 0.05$ |
| latent size | $64$ |
| latent space MLP size | $128$ |
| decoder input channels | $66$ |
| number of skip connections in U-Net | $0, 3, 5$ |
| dataset | CLEVR, Multi-dSprites |

Table 1: Hyperparameter search for MONet.

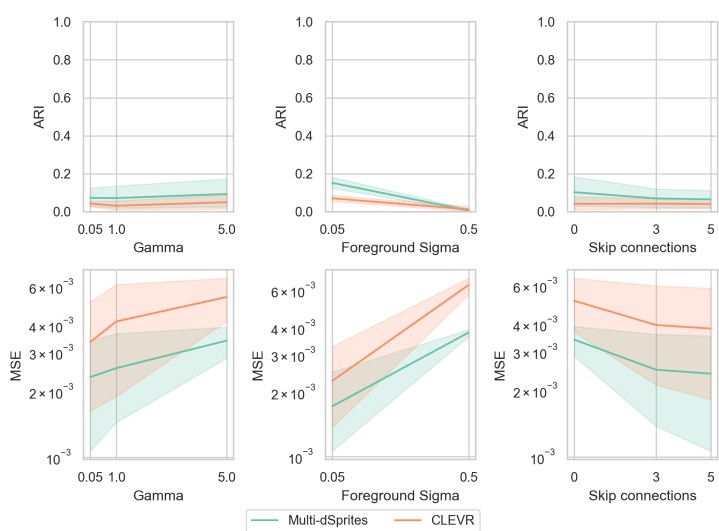

Figure 7: Hyperparameter search for MONet.

### E.3    REPRESENTATION BOTTLENECK STUDY

The representation bottleneck study was done by changing the latent space of both MONet and Slot Attention with 2 seeds and without changing any other parameter, resulting in 32 runs. The latent sizes tested are shown in Table 2. The findings are summarized in the main text of the paper.

| MONet | Slot Attention |
|---|---|
| 8 | 32 |
| 16 | 64 |
| 32 | 128 |
| 64 | 256 |
| 128 | 512 |

Table 2: Latent space sizes in the study.

### E.4    IMPROVING SLOT ATTENTION

We tried to increase the size of the encoder and decoder architecture as much as possible, while being reasonable regarding training time. We quickly realised that we needed a very deep architecture, therefore, we opted to use residual layers. Each layer is a stack of two convolutional layers,

with Leaky ReLU activation functions, a skip connection and we also employ the re-zero strategy (Bachlechner et al., 2021). We increased the latent size to 512, used upscaling in the encoder and downscaling in the decoder. We fixed the broadcast size of the decoder to 32. We used a stack of 16 residual blocks. The architecture of the encoder is described in Table 3, and the decoder is symmetrical (starting from 256 channels going down and instead of downscaling we have upscaling). To map from the input number of channels to the desired ones we use an additional convolutional layer, the same for the output channels. We did not experiment with the number of iterations that the Slot Attention Module performs, but it would be interesting to understand whether this parameter is very influential in natural scenes.

| Name | Number of channels | Activation/ Comment |
|---|---|---|
| Residual Block | 64 | Leaky ReLU |
| Residual Block | 64 | Leaky ReLU |
| Residual Block | 64 | Leaky ReLU |
| Residual Block | 64 | Leaky ReLU |
| Downscaling | | Only for CLEVR |
| Residual Block | 64 | Leaky ReLU |
| Residual Block | 64 | Leaky ReLU |
| Residual Block | 64 | Leaky ReLU |
| Residual Block | 64 | Leaky ReLU |
| Downscaling | | |
| Residual Block | 128 | Leaky ReLU |
| Residual Block | 128 | Leaky ReLU |
| Residual Block | 128 | Leaky ReLU |
| Residual Block | 128 | Leaky ReLU |
| Residual Block | 256 | Leaky ReLU |
| Residual Block | 256 | Leaky ReLU |
| Residual Block | 256 | Leaky ReLU |
| Residual Block | 256 | Leaky ReLU |

Table 3: Latent space sizes in the study.

# F    QUALITATIVE RESULTS

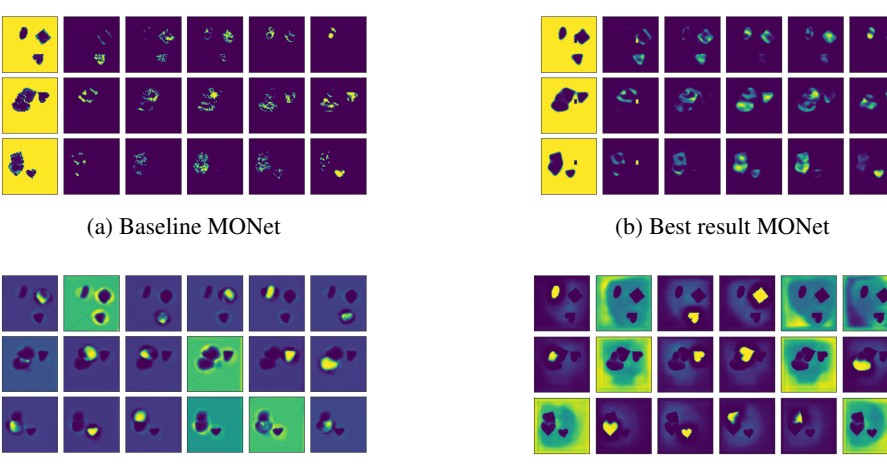

(a) Baseline MONet

(b) Best result MONet

(c) Baseline Slot Attention

(d) Best result Slot Attention

Figure 8: Qualitative results for the separation performance of the models in the comparative study on Multi-dSprites. From left to right in each subfigure: reconstruction and mask for each of the six slots. Here the difference between the two versions of Slot Attention is even more noticeable, and we can see how MONet is blurring the masks.

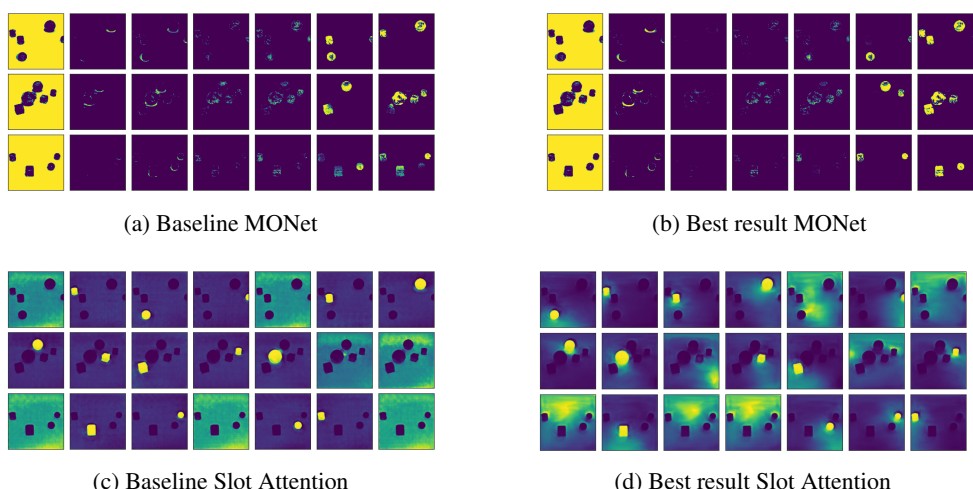

(a) Baseline MONet

(b) Best result MONet

(c) Baseline Slot Attention

(d) Best result Slot Attention

Figure 9: Qualitative results for the separation performance of the models in the comparative study on CLEVR. From left to right in each subfigure: reconstruction and mask for each of the seven slots. The masks on the improved Slot Attention start to include more of the background for each object. For MONet, it manages to perform better than Multi-dSprites, but the best result is still behaving in the same way.

