# OpenReview forum: "Inductive Biases for Object-Centric Representations of Complex Textures"
_ICLR.cc/2022/Workshop/OSC — Submitted to ICLR2022 OSC _

### Official Review · Reviewer_wSMN · 2022-03-14
**An interesting direction with some problems**

**Rating:** 1
**Confidence:** 2

**Review:**

This paper investigates the performance of two object centric representation models - MONet and Slot Attention on a generated dataset which adds textures to existing datasets (CLEVR and dSprites). This addition of textures makes the segmenation task harder and the paper investigates how the models cope with this change compared to the simple version of the dataset (which allows for example, segmenation by color to work quite well).

The main takeaway message from the paper is that models which reconstruct both shape and appearance together work better than ones that do the two separately. In the context of the paper this means that slot attention works better in these settings than MONet, even after fine tuning. I think I have two issues with the above claim - one is that there are many, many differences between the two models, some of them relate to how inference is done, some to how the generation is done. I don't think that it is possible to conclude the main paper's conclusion from the presented experiments. Furthermore, I don't even think it's true to say that MONet separates the shape from appearance on the representation side - afaik the generative model generates both apperanance *and* masks which are then softmaxed much like in slot-attention. It is true that the attention network produces these masks as targets as well (and separately!) but on the representation side I don't feel this is a significant difference. Beyond that, slot attention also "segments" the image separately while encoding the image - usually, examining the resulting attention pattern doesn't reveal much structure.

I think this set of experiments is nice and interesting, but it is a bit incomplete and does not support the claim made in the paper I'm afraid. I'm on the fence on acceptance here because in the context of a workshop these things are usually helpful to discuss - I'm willing to change my mind if other reviewers think that the problems I have presented here are not enough for rejection.

---

### Official Review · Reviewer_MpjW · 2022-03-16

**Rating:** 2
**Confidence:** 3

**Review:**

This paper explores the behavior of MONet and Slot Attention on controlled datasets with non-trivial textures generated by applying some neural style transfer to Multi-dSprites and CLEVR.
They find that MONet has a tendency to over segment (using color as segmentation cue and not the whole object with a given texture), while Slot Attention is better behaved.


Overall, this is a simple but clean paper which clearly exposes an issue that many practitioners have been aware of for some time but would benefit from being clearly communicated to the community. The datasets would also be interesting contributions as they lie at an interesting intersection of complexity when having access to groundtruth factor is important but where random textures are considered valuable.


Questions and comments:
1. The exact way neural style transfer was applied wasn’t the most clearly explained. AFAIK, the whole image is being modified, and the segments are only used afterwards to undo the changes to the background.
   1. You could have used the segments to apply different random textures “per object” and combine them back again?
   2. Especially for CLEVR, the textures aren’t particularly visually apparent in the current version, would this have helped?
2. One usual way to try to make MONet not over segment is to increase Beta to increase the KL pressure towards the Normal prior.
   1. Did you try this by any chance? (Note that it is still not easy to avoid the color-segmentation behavior even when doing this)
   2. You swept “gamma”’. Just to make sure, this is the gamma that controls the pressure to reconstruct the masks in Equation 3 in the MONet paper, correct?
3. Instead of Figure 4 with Pearson coefficients, I’d personally prefer to see the raw scatter plots and assess the correlation visually myself. This might be nice to include in the Appendix.
4. Nit: the last sentence of the abstract is a bit too terse and only makes sense after having read the paper. I would try to reword it to be more explicit and provide more context.

---

### Decision · Program_Chairs · 2022-03-23

**Decision:**

Reject

**Comment:**

I agree with the concerns of reviewer wSMN. The experiments (while interesting and worth sharing!), do not support the main claim of the paper. Contrary to the claim in the paper, MONet indeed does reconstruct both shape and appearance from its representation, though it is true that shape inference is handled very differently. However, as wSMN points out, that is just one of many differences and the experiments presented in the paper do not sufficiently isolate this aspect. This concern weighs even stronger, in light of the recent ClevrTex paper[1] that runs a somewhat similar investigation but yields different results.
I don't think this can be addressed by a minor revision, so I'll reject the paper in its current form, but encourage the authors to extend/rewrite the paper and share their results with the larger community.

[1] Karazija, L., Laina, I., & Rupprecht, C. (2021). Clevrtex: A texture-rich benchmark for unsupervised multi-object segmentation. arXiv preprint arXiv:2111.10265.